# Modeling Overlapping Communities with Node Popularities

**Prem Gopalan**[1], **Chong Wang**[2], and **David M. Blei**[1]

[1]Department of Computer Science, Princeton University, {pgopalan,blei}@cs.princeton.edu
[2]Machine Learning Department, Carnegie Mellon University, {chongw}@cs.cmu.edu

## Abstract

We develop a probabilistic approach for accurate network modeling using node popularities within the framework of the mixed-membership stochastic blockmodel (MMSB). Our model integrates two basic properties of nodes in social networks: homophily and preferential connection to popular nodes. We develop a scalable algorithm for posterior inference, based on a novel nonconjugate variant of stochastic variational inference. We evaluate the link prediction accuracy of our algorithm on nine real-world networks with up to 60,000 nodes, and on simulated networks with degree distributions that follow a power law. We demonstrate that the AMP predicts significantly better than the MMSB.

## 1   Introduction

Social network analysis is vital to understanding and predicting interactions between network entities [6, 19, 21]. Examples of such networks include online social networks, collaboration networks and hyperlinked blogs. A central problem in social network analysis is to identify hidden community structures and node properties that can best explain the network data and predict connections [19].

Two node properties underlie the most successful models that explain how network connections are generated. The first property is *popularity*. This is the basis for preferential attachment [12], according to which nodes preferentially connect to popular nodes. The resulting degree distributions from this process are known to satisfy empirically observed properties such as power laws [24]. The second property that underlies many network models is *homophily* or *similarity*, according to which nodes with similar observed or unobserved attributes are more likely to connect to each other. To best explain social network data, a probabilistic model must capture these competing node properties.

Recent theoretical work [24] has argued that optimizing the trade-offs between popularity and similarity best explains the evolution of many real networks. It is intuitive that combining both notions of attractiveness, i.e., popularity and similarity, is essential to explain how networks are generated. For example, on the Internet a user's web page may link to another user due to a common interest in skydiving. The same user's page may also link to popular web pages such as Google.com.

In this paper, we develop a probabilistic model of networks that captures both popularity and homophily. To capture homophily, our model is built on the mixed-membership stochastic blockmodel (MMSB) [2], a community detection model that allows nodes to belong to multiple communities. (For example, a member of a large social network might belong to overlapping communities of neighbors, co-workers, and school friends.) The MMSB provides better fits to real network data than single community models [23, 27], but cannot account for node popularities.

Specifically, we extend the assortative MMSB [9] to incorporate per-community node popularity. We develop a scalable algorithm for posterior inference, based on a novel nonconjugate variant of stochastic variational inference [11]. We demonstrate that our model predicts significantly better

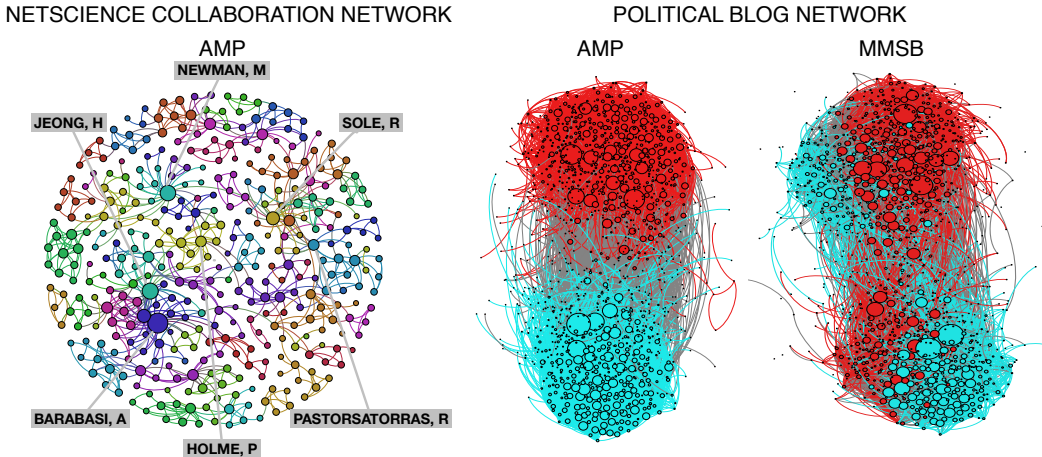

Figure 1: We visualize the discovered community structure and node popularities in a giant component of the netscience collaboration network [22] (Left). Each link denotes a collaboration between two authors, colored by the posterior estimate of its community assignment. Each author node is sized by its estimated posterior popularity and colored by its dominant research community. The network is visualized using the Fructerman-Reingold algorithm [7]. Following [14], we show an example where incorporating node popularities helps in accurately identifying communities (Right). The division of the political blog network [1] discovered by the AMP corresponds closely to the liberal and conservative blogs identified in [1]; the MMSB has difficulty in delineating these groups.

than the stochastic variational inference algorithm for the MMSB [9] on nine large real-world networks. Further, using simulated networks, we show that node popularities are essential for predictive accuracy in the presence of power-law distributed node degrees.

**Related work.** There have been several research efforts to incorporate popularity into network models. Karrer et al. [14] proposed the degree-corrected blockmodel that extends the classic stochastic blockmodels [23] to incorporate node popularities. Krivitsky et al. [16] proposed the latent cluster random effects model that extends the latent space model [10] to include node popularities. Both models capture node similarity and popularity, but assume that unobserved similarity arises from each node participating in a single community. Finally, the Poisson community model [4] is a probabilistic model of overlapping communities that implicitly captures degree-corrected mixed-memberships. However, the standard EM inference under this model drives many of the per-node community parameters to zero, which makes it ineffective for prediction or model metrics based on prediction (e.g., to select the number of communities).

## 2 Modeling node popularity and similarity

The assortative mixed-membership stochastic blockmodel (MMSB) [9] treats the links or non-links $y_{ab}$ of a network as arising from interactions between nodes $a$ and $b$. Each node $a$ is associated with *community memberships* $\pi_a$, a distribution over communities. The probability that two nodes are linked is governed by the similarity of their community memberships and the strength of their shared communities.

Given the communities of a pair of nodes, the link indicators $y_{ab}$ are independent. We draw $y_{ab}$ repeatedly by choosing a community assignment $(z_{a \to b}, z_{a \leftarrow b})$ for a pair of nodes $(a, b)$, and drawing a binary value from a community distribution. Specifically, the conditional probability of a link in MMSB is

$$p(y_{ab} = 1 | z_{a \to b, i}, z_{a \leftarrow b, j}, \boldsymbol{\beta}) = \sum_{i=1}^{K} \sum_{j=1}^{K} z_{a \to b, i} z_{a \leftarrow b, j} \beta_{ij},$$

where $\boldsymbol{\beta}$ is the blockmodel matrix of community strength parameters to be estimated. In the assortative MMSB [9], the non-diagonal entries of the blockmodel matrix are set close to 0. This captures node similarity in community memberships—if two nodes are linked, it is likely that the latent community indicators were the same.

In the proposed model, *assortative MMSB with node popularities*, or AMP, we introduce latent variables $\theta_a$ to capture the popularity of each node $a$, i.e., its propensity to attract links independent of its community memberships. We capture the effect of node popularity and community similarity on link generation using a logit model

$$\text{logit}\left(p(y_{ab} = 1 | z_{a \to b}, z_{a \leftarrow b}, \boldsymbol{\theta}, \boldsymbol{\beta})\right) \equiv \theta_a + \theta_b + \sum_{k=1}^{K} \delta_{ab}^k \beta_k, \tag{1}$$

where we define indicators $\delta_{ab}^k = z_{a \to b, k} z_{a \leftarrow b, k}$. The indicator $\delta_{ab}^k$ is one if both nodes assume the same community $k$.

Eq. 1 is a log-linear model [20]. In log-linear models, the random component, i.e., the expected probability of a link, has a multiplicative dependency on the systematic components, i.e., the covariates. This model is also similar in the spirit of the random effects model [10]—the node-specific effect $\theta_a$ captures the popularity of individual nodes while the $\sum_{k=1}^{K} \delta_{ab}^k \beta_k$ term captures the interactions through latent communities. Notice that we can easily extend the predictor in Eq. 1 to include observed node covariates, if any.

We now define a hierarchical generative process for the observed link or non-link under the AMP:

1. Draw $K$ community strengths $\beta_k \sim \mathcal{N}(\mu_0, \sigma_0^2)$.
2. For each node $a$,
   (a) Draw community memberships $\pi_a \sim \text{Dirichlet}(\alpha)$.
   (b) Draw popularity $\theta_a \sim \mathcal{N}(0, \sigma_1^2)$.
3. For each pair of nodes $a$ and $b$,
   (a) Draw interaction indicator $z_{a \to b} \sim \pi_a$.
   (b) Draw interaction indicator $z_{a \leftarrow b} \sim \pi_b$.
   (c) Draw the probability of a link $y_{ab} | z_{a \to b}, z_{a \leftarrow b}, \theta, \beta \sim \text{logit}^{-1}(z_{a \to b}, z_{a \leftarrow b}, \theta, \beta)$.

Under the AMP, the similarities between the nodes' community memberships and their respective popularities compete to explain the observations.

We can make AMP simpler by replacing the vector of $K$ latent community strengths $\boldsymbol{\beta}$ with a single community strength $\beta$. In §4, we demonstrate that this simpler model gives good predictive performance on small networks.

We analyze data with the AMP via the posterior distribution over the latent variables $p(\pi_{1:N}, \theta_{1:N}, \boldsymbol{z}, \beta_{1:K} | \boldsymbol{y}, \alpha, \mu_0, \sigma_0^2, \sigma_1^2)$, where $\theta_{1:N}$ represents the node popularities, and the posterior over $\pi_{1:N}$ represents the community memberships of the nodes. With an estimate of this latent structure, we can characterize the network in many useful ways. Figure 1 gives an example.

This is a subgraph of the netscience collaboration network [22] with $N = 1460$ nodes. We analyzed this network with $K = 100$ communities, using the algorithm from §3. This results in posterior estimates of the community memberships and popularities for each node and posterior estimates of the community assignments for each link. With these estimates, we visualized the discovered community structure and the popular authors.

In general, with an estimate of this latent structure, we can study individual links, characterizing the extent to which they occur due to similarity between nodes and the extent to which they are an artifact of the popularity of the nodes.

## 3 A stochastic gradient algorithm for nonconjugate variational inference

Our goal is to compute the posterior distribution $p(\pi_{1:N}, \theta_{1:N}, \boldsymbol{z}, \beta_{1:K} | \boldsymbol{y}, \alpha, \mu_0 \sigma_0^2, \sigma_1^2)$. Exact inference is intractable; we use variational inference [13].

Traditionally, variational inference is a coordinate ascent algorithm. However, the AMP presents two challenges. First, in variational inference the coordinate updates are available in closed form only when all the nodes in the graphical model satisfy conditional conjugacy. The AMP is not conditionally conjugate. To see this, note that the Gaussian priors on the popularity $\theta$ and the community strengths $\beta$ are not conjugate to the conditional likelihood of the data. Second, coordinate ascent algorithms iterate over all the $O(N^2)$ node pairs making inference intractable for large networks.

We address these challenges by deriving a stochastic gradient algorithm that optimizes a tractable lower bound of the variational objective [11]. Our algorithm avoids the $O(N^2)$ computational cost per iteration by subsampling a "mini-batch" of random nodes and a subset of their interactions in each iteration [9].

### 3.1 The variational objective

In variational inference, we define a family of distributions over the hidden variables $q(\boldsymbol{\beta}, \boldsymbol{\theta}, \boldsymbol{\pi}, \boldsymbol{z})$ and find the member of that family that is closest to the true posterior. We use the mean-field family, with the following variational distributions:

$$q(z_{a \to b} = i, z_{a \leftarrow b} = j) = \phi_{ab}^{ij}; \quad q(\pi_n) = \text{Dirichlet}(\pi_n; \gamma_n);$$
$$q(\beta_k) = \mathcal{N}(\beta_k; \mu_k, \sigma_\beta^2); \qquad q(\theta_n) = \mathcal{N}(\theta_n; \lambda_n, \sigma_\theta^2). \tag{2}$$

The posterior over the joint distribution of link community assignments per node pair $(a, b)$ is parameterized by the *per-interaction memberships* $\phi_{ab}$ [1], the community memberships by $\gamma$, the community strength distributions by $\mu$ and the popularity distributions by $\lambda$.

Minimizing the KL divergence between $q$ and the true posterior is equivalent to optimizing an *evidence lower bound* (ELBO) $\mathcal{L}$, a bound on the log likelihood of the observations. We obtain this bound by applying Jensen's inequality [13] to the data likelihood. The ELBO is

$$
\begin{aligned}
\mathcal{L} &= \sum_n \mathbb{E}_q[\log p(\pi_n|\alpha)] - \sum_n \mathbb{E}_q[\log q(\pi_n|\gamma_n)] \\
&+ \sum_n \mathbb{E}_q[\log p(\theta_n|\sigma_1^2)] - \sum_n \mathbb{E}_q[\log q(\theta_n|\lambda_n, \sigma_\theta^2)] \\
&+ \sum_k \mathbb{E}_q[\log p(\beta_k|\mu_0, \sigma_0^2)] - \sum_k \mathbb{E}_q[\log q(\beta_k|\mu_k, \sigma_\beta^2)] \\
&+ \sum_{a,b} \mathbb{E}_q[\log p(z_{a \to b}|\pi_a)] + \mathbb{E}_q[\log p(z_{a \leftarrow b}|\pi_b)] - \mathbb{E}_q[\log q(z_{a \to b}, z_{a \leftarrow b}|\phi_{ab})] \\
&+ \sum_{a,b} \mathbb{E}_q[\log p(y_{ab}|z_{a \to b}, z_{a \leftarrow b}, \boldsymbol{\theta}, \boldsymbol{\beta})]. 
\end{aligned} \tag{3}
$$

Notice that the first three lines in Eq. 3 contains summations over communities and nodes; we call these *global terms*. They relate to the global parameters which are $(\boldsymbol{\gamma}, \boldsymbol{\lambda}, \boldsymbol{\mu})$. The remaining lines contain summations over all node pairs; we call these *local terms*. They relate to the local parameters which are the $\phi_{ab}$. The distinction between the global and local parameters is important—the updates to global parameters depends on all (or many) local parameters, while the updates to local parameters for a pair of nodes only depends on the relevant global and local parameters in that context.

Estimating the global variational parameters is a challenging computational problem. Coordinate ascent inference must consider each pair of nodes at each iteration, but even a single pass through the $O(N^2)$ node pairs can be prohibitive. Previous work [9] has taken advantage of conditional conjugacy of the MMSB to develop fast stochastic variational inference algorithms. Unlike the MMSB, the AMP is not conditionally conjugate. Nevertheless, by carefully manipulating the variational objective, we can develop a scalable stochastic variational inference algorithm for the AMP.

### 3.2 Lower bounding the variational objective

To optimize the ELBO with respect to the local and global parameters we need its derivatives. The data likelihood terms in the ELBO can be written as

$$\mathbb{E}_q[\log p(y_{ab}|z_{a \to b}, z_{a \leftarrow b}, \boldsymbol{\theta}, \boldsymbol{\beta})] = y_{ab}\mathbb{E}_q[x_{ab}] - \mathbb{E}_q[\log(1 + \exp(x_{ab}))], \tag{4}$$

where we define $x_{ab} \equiv \theta_a + \theta_b + \sum_{k=1}^K \beta_k \delta_{ab}^k$. The terms in Eq. 4 cannot be expanded analytically. To address this issue, we further lower bound $-\mathbb{E}_q[\log(1+\exp(x_{ab}))]$ using Jensen's inequality [13],

$$
\begin{aligned}
-\mathbb{E}_q[\log(1 + \exp(x_{ab}))] &\geq -\log[\mathbb{E}_q(1 + \exp(x_{ab}))] \\
&= -\log[1 + \mathbb{E}_q[\exp(\theta_a + \theta_b + \sum_{k=1}^K \beta_k \delta_{ab}^k)]] \\
&= -\log[1 + \exp(\lambda_a + \sigma_\theta^2/2)\exp(\lambda_b + \sigma_\theta^2/2)s_{ab}], 
\end{aligned} \tag{5}
$$

**Algorithm 1** The stochastic AMP algorithm

---
1: Initialize variational parameters. See §3.5.
2: **while** convergence criteria is not met **do**
3:     Sample a mini-batch $S$ of nodes. Let $P$ be the set of node pairs in $S$.
4:     **local step**
5:     Optimize $\phi_{ab} \; \forall (a, b) \in P$ using Eq. 11 and Eq. 12.
6:     **global step**
7:     Update memberships $\gamma_a$, for each node $a \in S$, using stochastic natural gradients in Eq. 6.
8:     Update popularities $\lambda_a$, for each node $a \in S$ using stochastic gradients in Eq. 7.
9:     Update community strengths $\boldsymbol{\mu}$ using stochastic gradients in Eq. 9.
10:     Set $\rho_a(t) = (\tau_0 + t_a)^{-\kappa}; t_a \leftarrow t_a + 1$, for each node $a \in S$.
11:     Set $\rho'(t) = (\tau_0 + t)^{-\kappa}; t \leftarrow t + 1$.
12: **end while**

---

where we define $s_{ab} \equiv \sum_{k=1}^{K} \phi_{ab}^{kk} \exp\{\mu_k + \sigma_\beta^2/2\} + (1 - \sum_{k=1}^{K} \phi_{ab}^{kk})$. In simplifying Eq. 5, we have used that $q(\theta_n)$ is a Gaussian. Using the mean of a log-normal distribution, we have $\mathbb{E}_q[\exp(\theta_n)] = \exp(\lambda_n + \sigma_\theta^2/2)$. A similar substitution applies for the terms involving $\beta_k$ in Eq. 5.

We substitute Eq. 5 in Eq. 3 to obtain a tractable lower bound $\mathcal{L}'$ of the ELBO $\mathcal{L}$ in Eq. 3. This allows us to develop a coordinate ascent algorithm that iteratively updates the local and global parameters to optimize this lower bound on the ELBO.

## 3.3 The global step

We optimize the ELBO with respect to the global variational parameters using stochastic gradient ascent. Stochastic gradient algorithms follow noisy estimates of the gradient with a decreasing step-size. If the expectation of the noisy gradient equals to the gradient and if the step-size decreases according to a certain schedule, then the algorithm converges to a local optimum [26]. Subsampling the data to form noisy gradients scales inference as we avoid the expensive all-pairs sums in Eq. 3.

The global step updates the global community memberships $\boldsymbol{\gamma}$, the global popularity parameters $\boldsymbol{\lambda}$ and the global community strength parameters $\boldsymbol{\mu}$ with a stochastic gradient of the lower bound on the ELBO $\mathcal{L}'$. In [9], the authors update community memberships of all nodes after each iteration by obtaining the natural gradients of the ELBO [2] with respect to the vector $\boldsymbol{\gamma}$ of dimension $N \times K$. We use natural gradients for the memberships too, but use distinct stochastic optimizations for the memberships and popularity parameters of each node and maintain a separate learning rate for each node. This restricts the per-iteration updates to nodes in the current mini-batch.

Since the variational objective is a sum of terms, we can cheaply compute a stochastic gradient by first subsampling a subset of terms and then forming an appropriately scaled gradient. We use a variant of the random node sampling method proposed in [9]. At each iteration we sample a node uniformly at random from the $N$ nodes in the network. (In practice we sample a "mini-batch" $S$ of nodes per update to reduce noise [11, 9].) While a naive method will include all interactions of a sampled node as the observed pairs, we can leverage network sparsity for efficiency; in many real networks, only a small fraction of the node pairs are linked. Therefore, for each sampled node, we include as observations all of its links and a small uniform sample of $m_0$ non-links.

Let $\partial \gamma_a^t$ be the natural gradient of $\mathcal{L}'$ with respect to $\gamma_a$, and $\partial \lambda_a^t$ and $\partial \mu_k^t$ be the gradients of $\mathcal{L}'$ with respect to $\lambda_a$ and $\mu_k$, respectively. Following [2, 9], we have

$$\partial \gamma_{a,k}^t = -\gamma_{a,k}^{t-1} + \alpha_k + \sum_{(a,b) \in \text{links}(a)} \phi_{ab}^{kk}(t) + \sum_{(a,b) \in \text{nonlinks}(a)} \phi_{ab}^{kk}(t), \quad (6)$$

where links(a) and nonlinks(a) correspond to the set of links and non-links of $a$ in the training set. Notice that an unbiased estimate of the summation term over non-links in Eq. 6 can be obtained from a subsample of the node's non-links. Therefore, the gradient of $\mathcal{L}'$ with respect to the membership parameter $\gamma_a$, computed using all of the nodes' links and a subsample of its non-links, is a noisy but unbiased estimate of the natural gradient in Eq. 6.

The gradient of the approximate ELBO with respect to the popularity parameter $\lambda_a$ is

$$\partial \lambda_a^t = -\frac{\lambda_a^{t-1}}{\sigma_1^2} + \sum_{(a,b)\in\text{links(a)} \cup \text{nonlinks(a)}}(y_{ab} - r_{ab}s_{ab}), \tag{7}$$

where we define $r_{ab}$ as

$$r_{ab} \equiv \frac{\exp\{\lambda_a + \sigma_\theta^2/2\}\exp\{\lambda_b + \sigma_\theta^2/2\}}{1 + \exp\{\lambda_a + \sigma_\theta^2/2\}\exp\{\lambda_b + \sigma_\theta^2/2\}s_{ab}}. \tag{8}$$

Finally, the stochastic gradient of $\mathcal{L}'$ with respect to the global community strength parameter $\mu_k$ is

$$\partial \mu_k^t = \frac{\mu_0 - \mu_k^{t-1}}{\sigma_0^2} + \frac{N}{2|S|}\sum_{(a,b)\in\text{links(S)} \cup \text{nonlinks(S)}}\phi_{ab}^{kk}(y_{ab} - r_{ab}\exp\{\mu_k + \sigma_\beta^2/2\}). \tag{9}$$

As with the community membership gradients, notice that an unbiased estimate of the summation term over non-links in Eq. 7 and Eq. 9 can be obtained from a subsample of the node's non-links. To obtain an unbiased estimate of the true gradient with respect to $\mu_k$, the summation over a node's links and non-links must be scaled by the inverse probability of subsampling that node in Eq. 9. Since each pair is shared between two nodes, and we use a mini-batch with $S$ nodes, the summations over the node pairs are scaled by $\frac{N}{2|S|}$ in Eq. 9.

We can interpret the gradients in Eq. 7 and Eq. 9 by studying the terms involving $r_{ab}$ in Eq. 7 and Eq. 9. In Eq. 7, $(y_{ab} - r_{ab}s_{ab})$ is the residual for the pair $(a,b)$, while in Eq. 9, $(y_{ab} - r_{ab}\exp\{\mu_k + \sigma_\beta^2/2\})$ is the residual for the pair $(a,b)$ conditional on the latent community assignment for both nodes $a$ and $b$ being set to $k$. Further, notice that the updates for the global parameters of node $a$ and $b$, and the updates for $\boldsymbol{\mu}$ depend only on the diagonal entries of the indicator variational matrix $\phi_{ab}$.

We can similarly obtain stochastic gradients for the variational variances $\sigma_\beta$ and $\sigma_\theta$; however, in our experiments we found that fixing them already gives good results. (See §4.)

The global step for the global parameters follows the noisy gradient with an appropriate step-size:

$$\gamma_a \leftarrow \gamma_a + \rho_a(t)\partial\gamma_a^t; \quad \lambda_a \leftarrow \lambda_a + \rho_a(t)\partial\lambda_a^t; \quad \boldsymbol{\mu} \leftarrow \boldsymbol{\mu} + \rho'(t)\partial\boldsymbol{\mu}^t. \tag{10}$$

We maintain separate learning rates $\rho_a$ for each node $a$, and only update the $\gamma$ and $\lambda$ for the nodes in the mini-batch in each iteration. There is a global learning rate $\rho'$ for the community strength parameters $\boldsymbol{\mu}$, which are updated in every iteration. For each of these learning rates $\rho$, we require that $\sum_t \rho(t)^2 < \infty$ and $\sum_t \rho(t) = \infty$ for convergence to a local optimum [26]. We set $\rho(t) \triangleq (\tau_0 + t)^{-\kappa}$, where $\kappa \in (0.5, 1]$ is the learning rate and $\tau_0 \geq 0$ downweights early iterations.

### 3.4 The local step

We now derive the updates for the local parameters. The local step optimizes the per-interaction memberships $\phi$ with respect to a subsample of the network. There is a per-interaction variational parameter of dimension $K \times K$ for each node pair—$\phi_{ab}$—representing the posterior approximation of which pair of communities are active in determining the link or non-link. The coordinate ascent update for $\phi_{ab}$ is

$$\phi_{ab}^{kk} \propto \exp\left\{\mathbb{E}_q[\log\pi_{a,k}] + \mathbb{E}_q[\log\pi_{b,k}] + y_{ab}\mu_k - r_{ab}(\exp\{\mu_k + \sigma_\beta^2/2\} - 1)\right\} \tag{11}$$

$$\phi_{ab}^{ij} \propto \exp\left\{\mathbb{E}_q[\log\pi_{a,i}] + \mathbb{E}_q[\log\pi_{b,j}]\right\}, i \neq j, \tag{12}$$

where $r_{ab}$ is defined in Eq. 8. We present the full stochastic variational inference in Algorithm 1.

### 3.5 Initialization and convergence

We initialize the community memberships $\boldsymbol{\gamma}$ using approximate posterior memberships from the variational inference algorithm for the MMSB [9]. We initialized popularities $\boldsymbol{\lambda}$ to the logarithm of the normalized node degrees added to a small random offset, and initialized the strengths $\boldsymbol{\mu}$ to zero. We measure convergence by computing the link prediction accuracy on a validation set with 1% of the networks links, and an equal number of non-links. The algorithm stops either when the change in log-likelihood on this validation set is less than 0.0001%, or if the log-likelihood decreases for consecutive iterations.

Figure 2: Network data sets. $N$ is the number of nodes, $d$ is the percent of node pairs that are links and $P$ is the mean perplexity over the links and nonlinks in the held-out test set.

| DATA SET | $N$ | $d(\%)$ | $P_{\text{AMP}}$ | $P_{\text{MMSB}}$ | TYPE | SOURCE |
|---|---|---|---|---|---|---|
| US AIR | 712 | 1.7% | $\mathbf{2.75 \pm 0.04}$ | $3.41 \pm 0.15$ | TRANSPORT | [25] |
| POLITICAL BLOGS | 1224 | 1.9% | $\mathbf{2.97 \pm 0.03}$ | $3.12 \pm 0.01$ | HYPERLINK | [1] |
| NETSCIENCE | 1450 | 0.2% | $\mathbf{2.73 \pm 0.11}$ | $3.02 \pm 0.19$ | COLLAB. | [22] |
| RELATIVITY | 4158 | 0.1% | $\mathbf{3.69 \pm 0.18}$ | $6.53 \pm 0.37$ | COLLAB. | [18] |
| HEP-TH | 8638 | 0.05% | $\mathbf{12.35 \pm 0.17}$ | $23.06 \pm 0.87$ | COLLAB. | [18] |
| HEP-PH | 11204 | 0.16% | $\mathbf{2.75 \pm 0.06}$ | $3.310 \pm 0.15$ | COLLAB. | [18] |
| ASTRO-PH | 17903 | 0.11% | $\mathbf{5.04 \pm 0.02}$ | $5.28 \pm 0.07$ | COLLAB. | [18] |
| COND-MAT | 36458 | 0.02% | $\mathbf{10.82 \pm 0.09}$ | $13.52 \pm 0.21$ | COLLAB. | [22] |
| BRIGHTKITE | 56739 | 0.01% | $\mathbf{10.98 \pm 0.39}$ | $41.11 \pm 0.89$ | SOCIAL | [18] |

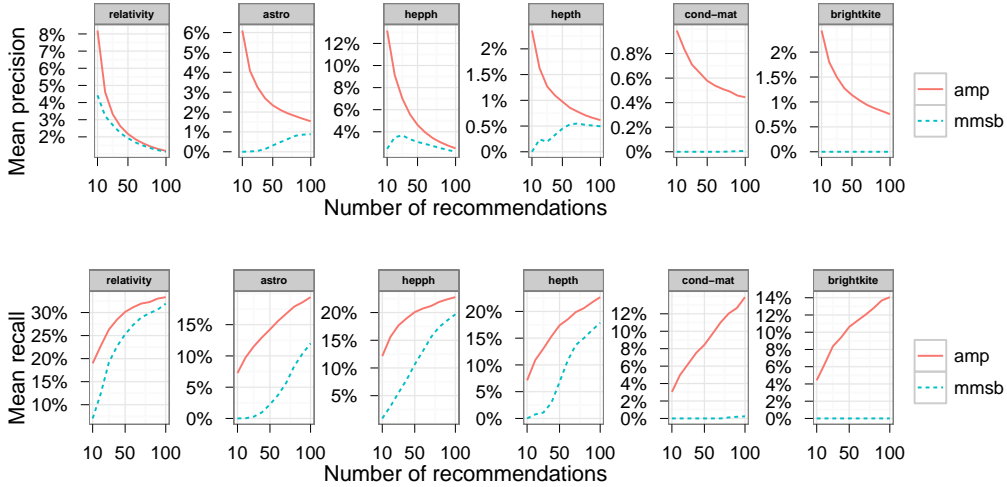

Figure 3: The AMP model outperforms the MMSB model of [9] in predictive accuracy on real networks. Both models were fit using stochastic variational inference [11]. For the data sets shown, the number of communities $K$ was set to 100 and hyperparameters were set to the same values across data sets. The perplexity results are based on five replications. A single replication is shown for the mean precision and mean recall.

## 4 Empirical study

We use the predictive approach to evaluating model fitness [8], comparing the predictive accuracy of AMP (Algorithm 1) to the stochastic variational inference algorithm for the MMSB with link sampling [9]. In all data sets, we found that AMP gave better fits to real-world networks. Our networks range in size from 712 nodes to 56,739 nodes. Some networks are sparse, having as little as 0.01% of all pairs as links, while others have up to 2% of all pairs as links. Our data sets contain four types of networks: hyperlink, transportation, collaboration and social networks. We implemented Algorithm 1 in 4,800 lines of C++ code. [3]

**Metrics.** We used perplexity, mean precision and mean recall in our experiments to evaluate the predictive accuracy of the algorithms. We computed the link prediction accuracy using a test set of node pairs that are not observed during training. The test set consists of 10% of randomly selected links and non-links from each data set. During training, these test set observations are treated as zeros. We approximate the predictive distribution of a held-out node pair $y_{ab}$ under the AMP using posterior estimates $\hat{\theta}$, $\hat{\beta}$ and $\hat{\pi}$ as

$$p(y_{ab}|\boldsymbol{y}) \approx \sum_{z_{a \to b}} \sum_{z_{a \leftarrow b}} p(y_{ab}|z_{a \to b}, z_{a \leftarrow b}, \hat{\theta}, \hat{\beta}) p(z_{a \to b}|\hat{\pi}_a) p(z_{a \leftarrow b}|\hat{\pi}_b). \tag{13}$$

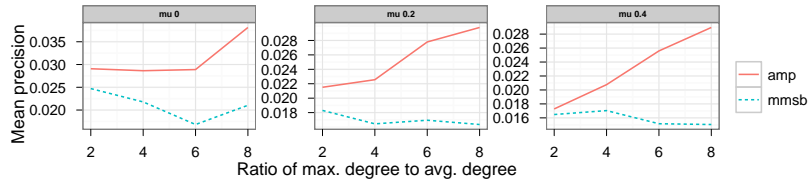

Figure 4: The AMP predicts significantly better than the MMSB [9] on 12 LFR benchmark networks [17]. Each plot shows 4 networks with increasing right-skewness in degree distribution. $\mu$ is the fraction of noisy links between dissimilar nodes—nodes that share no communities. The precision is computed at 50 recommendations for each node, and is averaged over all nodes in the network.

Perplexity is the exponential of the average predictive log likelihood of the held-out node pairs. For mean precision and recall, we generate the top $n$ pairs for each node ranked by the probability of a link between them. The ranked list of pairs for each node includes nodes in the test set, as well as nodes in the training set that were non-links. We compute precision-at-$m$, which measures the fraction of the top $m$ recommendations present in the test set; and we compute recall-at-$m$, which captures the fraction of nodes in the test set present in the top $m$ recommendations. We vary $m$ from 10 to 100. We then obtain the mean precision and recall across all nodes. [4]

**Hyperparameters and constants.** For the stochastic AMP algorithm, we set the "mini-batch" size $S = N/100$, where $N$ is the number of nodes in the network and we set the non-link sample size $m_0 = 100$. We set the number of communities $K = 2$ for the political blog network and $K = 20$ for the US air; for all other networks, $K$ was set to 100. We set the hyperparameters $\sigma_0^2 = 1.0$, $\sigma_1^2 = 10.0$ and $\mu_0 = 0$, fixed the variational variances at $\sigma_\theta = 0.1$ and $\sigma_\beta = 0.5$ and set the learning parameters $\tau_0 = 65536$ and $\kappa = 0.5$. We set the Dirichlet hyperparameter $\alpha = \frac{1}{K}$ for the AMP and the MMSB.

**Results on real networks.** Figure 2 compares the AMP and the MMSB stochastic algorithms on a number of real data sets. The AMP definitively outperforms the MMSB in predictive performance. All hyperparameter settings were held fixed across data sets. The first four networks are small in size, and were fit using the AMP model with a single community strength parameter. All other networks were fit with the AMP model with $K$ community strength parameters. As $N$ increases, the gap between the mean precision and mean recall performance of these algorithms appears to increase. Without node popularities, MMSB is dependent entirely on node memberships and community strengths to predict links. Since $K$ is held fixed, communities are likely to have more nodes as $N$ increases, making it increasingly difficult for the MMSB to predict links. For the small US air, political blogs and netscience data sets, we obtained similar performance for the replication shown in Figure 2. For the AMP the mean precision at 10 for US Air, political blogs and netscience were 0.087, 0.07, 0.092, respectively; for the MMSB the corresponding values were 0.007, 0.0, 0.063, respectively.

**Results on synthetic networks.** We generated 12 LFR benchmark networks [17], each with 1000 nodes. Roughly 50% of the nodes were assigned to 4 overlapping communities, and the other 50% were assigned to single communities. We set a community size range of [200, 500] and a mean node degree of 10 with power-law exponent set to 2.0. Figure 4 shows that the MMSB performs poorly as the skewness is increased, while the AMP performs significantly better in the presence of both noisy links and right-skewness, both characteristics of real networks. The skewness in degree distributions causes the community strength parameters of MMSB to overestimate or underestimate the linking patterns within communities. The per-node popularities in the AMP can capture the heterogeneity in node degrees, while learning the corrected community strengths.

**Acknowledgments**

David M. Blei is supported by ONR N00014-11-1-0651, NSF CAREER IIS-0745520, and the Alfred P. Sloan foundation. Chong Wang is supported by NSF DBI-0546594 and NIH 1R01GM093156.

## Footnotes

[1] Following [15], we use a structured mean-field assumption.

[2]The natural gradient [3] points in the direction of steepest ascent in the Riemannian space. The local distance in the Riemannian space is defined by KL divergence, a better measure of dissimilarity between probability distributions than Euclidean distance [11].

[3]Our software is available at https://github.com/premgopalan/sviamp.

[4]Precision and recall are better metrics than ROC AUC on highly skewed data sets [5].

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
