[Reviews · NeurIPS 2013]

Submitted by Assigned_Reviewer_4

This paper proposes a new generative model and associated link
inference method based on both node popularity and similarity. The
starting point for the model is the prior work in [11] where the
assortative mixed-membership stochastic blockmodel (AMMSB) was
presented. In the prior model, link structure is generated via community
strength (via a blockmodel) and community membership. In the new
work, link structure is generated by using the prior model and adding
"popularity" to the generative model.

After the model is presented, the authors then derive an optimization
criterion based upon a variational method (since exact inference is
impossible). The resulting criterion is optimized using a stochastic
gradient ascent technique.

The new model is then applied to real networks and the "standard" LFR
benchmarks for performance evaluation. Precision/recall curves show
that the new model is better at low-recall/high-precision operating
points. Additionally, Figure 5b shows a strong relation between node
popularity and node degree.

Quality

The paper is technically sound and the paper has extensive detail on
assumptions, methods, and algorithms. The results should be readily
reproducible.

Clarity

The paper could use improvement in the area of clarity. There is a
strong reliance on past work for notation and assumptions which make
this paper difficult to be read independently. A few comments:
- Some terms are not defined when they are first mentioned. E.g.,
(z_{a->b}, z_(b->a)) is called a community assignment (line 090). A simple
mention that this is a vector (or function on i=1,..,K) assuming
values 0 or 1 would be helpful. Also, \phi_{ab}^{ij) (line 160) is
never really defined.
- What is the relation between community strength, beta, in the new work and
in [11]? In the current work, beta is normal, but in the [11] it is a
Beta distribution.

Originality

This paper has a good result that builds off of previous work in
[11]. A comparison with [11] shows similarity in approach and
methodology. The methods are new, but they certainly build off of
prior work.

Significance
Results presented by the authors are significant. Methods to generate
and fit networks for link prediction continue to be a key task since
many real world networks are not observable. The fact that the
algorithm can produce high precision links (albeit with low recall) is
of interest in some applications. A drawback of the methods, possibly
for future work, is to continue to increase the scalability of the
method to larger networks.
Summary: This work is a solid new contribution for network modeling using a small parameter set. The authors present significant algorithms and results that demonstrate the effectiveness of their methods on real and simulated (LFR) networks over prior methods.

Submitted by Assigned_Reviewer_5

This paper proposes a novel model of social network data that learns latent structure in the network while adjusting for node popularity. The model is a straightforward extension of previous models - using a generalized linear model to include both popularity parameters that model the degree of each node as well as community memberships, akin to the MMSB.

The paper is a nice application of recent advances in stochastic variational inference. It would have been nice to see the reduction in time (in seconds) until convergence due to the stochastic approach (at least for small networks).

The paper is technically sound and the authors show that the AMP effectively reduces to AMMSB for synthetic data with unskewed degree distributions.

For each network it would be nice to better understand the role of epsilon and K in this model. Larger values of epsilon might allow for more between-community structure, which might be important for many networks. Also, are the results -- and the role of the popularity terms -- sensitive to the value of K?

It is also unclear how much of the predictive performance is due to the community structure after one includes the popularity terms. For example, in Figure 5b we see that the popularity estimates are quite sensible, but it would be nice to also see the estimates of beta (which are assumed to have been used to color Figure 5a).

The paper is clear and enjoyable to read.

Aside:
I am a bit confused about Figure 2. Is there something missing? And in the Table, why not include the mean AUC for other methods?
Summary: The paper represents a fine addition to the scalable modeling of social network data. The paper could benefit from a more thorough evaluation of the benefits of the approach.

Submitted by Assigned_Reviewer_6

The paper proposes a modification of Mixed Membership Stochastic Blockmodels (MMSB) to include node offset terms, which allows "popular" nodes to be more frequently involved in the formation of edges, regardless of their latent community memberships.

Overally, this is a nicely written paper, and it is well fleshed-out technically.

Cynically, the modification proposed compared to (A)MMSB is actually quite incremental, it essentially just adds some bias terms. Surprisingly, in spite of this minor modification, a huge amount of machinery has to be proposed by the authors to fit their new model, which in the end takes up nearly half the paper.

Given recent work (e.g. [11]) on scaling-up MMSB, it was a shame that this inference procedure couldn't have been presented more simply. At times in Section 3 I was uncertain what ideas were taken from [11] and what was new. Overall I would have liked to see the variational inference procedure presented more concisely (or moved to supp. material), and more space devoted to experiments/discussion.

The experiments were not bad, but again I'm disappointed by how little space is given to them.

You should say a bit more about why you chose these 8 networks in Figure 2. Since the algorithms being compared require only adjacency matrices, there are hundreds of networks that *could* be used, and it's easy to cherry-pick with this type of paper.

Confusingly, Figures 2 and 3 describe different networks (Condmat is missing from Figure 2, and US-air, netsci, and brightkite are missing from Figure 3). Please try and be consistent here. Although Figure 2 shows R_AMP for these datasets, they're never compared with other methods, so I have no clue whether R_AMP=0.8 is "good" for brightkite.

I didn't understand why the PR curves of AMP and AMMSB ought to line up when recall is high. There must be a mathematical explanation for this. My guess is that AMP is much more accurate at predicting edges incident on high-popularity nodes, and no more accurate elsewhere, but you should explain this, whatever the reason.

Since the models you're comparing are all probabilistic, why not report the perplexity on all datasets? From the plots shown, I can see that you clearly win on "gravity" and "hep-ph", but the benefits aren't clear elsewhere.

Figure 5 doesn't seem to be referred to anywhere in the paper.
Summary: A nicely written and technically solid paper. However, in spite of the considerable technical depth, the "meat" of the contribution is a bit incremental. The experimental section isn't given enough space and seems to skip some critical details.
Author Feedback

Author rebuttal: Thank you for your constructive reviews and suggestions. We've addressed your questions and concerns below.

1. R6 noted that the modification proposed to AMMSB [3] is minor and that it just adds bias terms. Our model (AMP) presents a significant improvement over the AMMSB. We've introduced latent variables for node popularities and a generalized linear model response.

These modifications make the AMP a nonconjugate model and therefore challenging to fit to massive networks. In addition to devising and studying the model, our contribution is a novel nonconjugate variant of stochastic variational inference for scalable posterior inference under the AMP. By capturing node popularities, we demonstrate significant improvements in predictive performance relative to the AMMSB. Our contributions include experimental validation on real networks with up to 57,000 nodes, and a study on synthetic benchmark networks with skewed degree distributions and significant mixing.

2. R6 asked to report on the perplexity. We show the perplexity of the AMMSB and the AMP on all datasets below. The best algorithm is marked by stars. (Note the Poisson model cannot compute perplexity due to zero-valued parameters [2].) We will report this in the paper.

The AMP has a significantly lower perplexity than the AMMSB on all datasets.

(The held-out set consists of a large number of non-links. Therefore, the perplexity scores on the combined set of links and non-links are small in magnitude. Both models can predict most non-links with high accuracy.)

----------Perplexity Table-----------------
Dataset | Method | Links | Links & non-links
----------------------------------------
astro-ph | AMP | 29.96 * | 1.006 *
astro-ph | AMMSB | 46.99 | 1.007

hep-ph | AMP | 7.17 * | 1.009 *
hep-ph | AMMSB | 14.88 | 1.012

brightkite | AMP | 275.89 * | 1.002 *
brightkite | AMMSB | 4298.40 | 1.003

hep-th | AMP | 68.03 * | 1.009 *
hep-th | AMMSB | 502.70 | 1.012

gravity | AMP | 13.73 * | 1.013 *
gravity | AMMSB | 48.90 | 1.018

netscience | AMP | 11.13 * | 1.029 *
netscience | AMMSB | 38.47 | 1.033

usair | AMP | 8.39 * | 1.099 *
usair | AMMSB | 13.28 | 1.116
----------------------------------------

3. R5 asked if the results are sensitive to the number of communities. Our key result is that the AMP has better precision-recall (PR) and predictive performance than the AMMSB on all datasets. This result is not sensitive to the number of communities; in our experiments, both models are set to the same number of communities and across numbers of communities, AMP had better PR curves.

4. R6 asked why we picked these datasets. We presented a systematic study of large, publicly accessible, collaboration, transportation and social networks. We did not cherry-pick these datasets.

5. R6 noted that "condmat" dataset is missing in Fig. 2. This is a typo. The "condmat" plot in Fig. 3 should be labeled "brightkite".

6. R6 commented on why the PR curves of AMP and AMMSB line up. R6's insight is correct. AMP sees advantage in predicting links influenced by popularity terms. When links are explained due to node similarity in communities, the AMMSB and the AMP PR curves line up.

7. R5 and R6 noted that we show the mean ROC AUC scores only for the AMP model. On highly-skewed datasets, such as network data, ROC AUC scores are misleading [1]. We will remove the ROC column from Fig. 2. We evaluate performance using Precision-Recall curves and Perplexity.

8. R5 asked how much of the predictive performance is due to node popularities. Please see our perplexity results in (2) above. The AMP's major improvement over the AMMSB is due to modeling node popularities. We will include estimates of the community strengths in our exploratory analysis.

9. R5 asked for the reduction in convergence time due to stochastic inference. This has been studied for the AMMSB [3]. For the AMP, the standard batch variational inference does not exist because the model is not conditionally conjugate.

10. R4 asked about the relation between community strength (\beta) in AMP and AMMSB. In AMP, the exp(\beta_k) is the effect of community k on the odds ratio given that the nodes assume community k in an interaction. In AMMSB, the \beta_k is the conditional probability of a link given that the nodes assume community k in an interaction.

11. R2 and R3 noted the missing usair, netscience PR curve plots. We omitted showing the PR curves on these small networks due to space constraints. We will add them to the supplement. In both plots, the AMP has a better performance than the AMMSB.

References

[1] J. Davis and M. Goadrich. The relationship between precision-recall and ROC curves.

[2] B. Ball, B. Karrer, MEJ Neman. Efficient and principled method for detecting communities in networks.

[3] P. Gopalan, D. Mimno, S. Gerrish, M. Freedman, and D. Blei. Scalable inference of overlapping communities.